# Enantiopure synthesis of [5]helicene based molecular lemniscates and their use in chiroptical materials

Leah E. M. White[1], Tiberiu-M. Gianga [2], Fabienne Pradaux-Caggiano[1], Chiara Faverio [1], Andrea Taddeucci [2,3], Henry S. Rzepa[4], Christian Jonhannesen [5], Lauren E. Hatcher[6], Giuliano Siligardi [2], David R. Carbery[1] & G. Dan Pantoș [1] ✉

The ability to synthesise lemniscular molecules to allow for the study and application of their chiroptical properties is a notable technical challenge. Herein, we report the design and synthesis of enantiomers of a [5]helicenoid derived molecular lemniscate, in which two homochiral helicenes are linked via the formation of two azine motifs. We demonstrate that these molecules, and their helicenoid constituents, are also excellent chiral dopants that induce dissymmetry in the ground and excited states of the achiral emissive polymer F8BT, leading to high CPL activity. The ability to control the handedness of the helicenoid dopants via enantiopure synthesis affords control of the sign of CP emission. This manipulation of circularly polarised light is of great interest for optoelectronic technologies.

The designed synthesis of uniquely shaped three-dimensional chiral molecules and molecular assemblies is an area of challenging and rewarding synthetic research. Molecular lemniscates, that is, molecules with a figure-eight shape can be included in this discussion and, in recent years, have gained interest among both synthetic and theoretical chemists. In a molecular context, the figure-eight shape not only results in visually impactful structures but also has embedded chirality[1].

Arguably, expanded porphyrins are the best-known examples of molecular structures shown to adopt figure-eight geometries[2–4]; however, more recently, bispyrrolidinoindoline-based macrocycles[5], nanohoops[6], nanobelts[7–9] and helicene dimers[10–14] have all been reported to demonstrate configurationally stable figure-eight structures. In addition to their syntheses, the chemical and photophysical properties of these uniquely shaped molecules have also been explored. As a result of their inherent chirality, molecular lemniscates have been investigated particularly for their chiroptical properties, notably in recent years for their ability to emit circularly polarised luminescence (CPL)[12,15]. Although limited examples of helicene-derived lemniscates have been reported to date, their design and synthesis can benefit from the more established

field of helicene chemistry[16–19]. The diverse array of helicene building blocks and synthetic procedures should enable easier access to these unique structures, allowing for further investigation of their properties. Indeed, previously reported work[11,12] utilises the popular photo-cyclodehydrogenation strategy for helicene formation in the lemniscate construction. Additionally, the synthesis of a figure-eight helicene dimer derived from a more structurally flexible expanded helicene scaffold highlights the diversity of helicene building blocks that may be used in their formation[13]. Despite this, all of the examples of helicene-derived lemniscates known to date are obtained either as the racemic mixture or undergo an optical resolution within the synthetic route in order to isolate the enantiopure molecules.

Organic chiral semiconductors have emerged as prospective materials with broad applications in optical data storage, bio-responsive imaging systems as well as next-generation displays or circularly polarised organic light-emitting diodes (CP-OLED)[20–23]. One class of molecules that have been used in CP-OLED applications are chiral conjugated polymers. These materials can be obtained either through the polymerisation of chiral monomers or by addition of small-molecule chiral

[1]Department of Chemistry, University of Bath, Bath, UK. [2]B23 Beamline, Diamond Light Source Ltd., Didcot, UK. [3]Department of Chemistry and Industrial Chemistry, University of Pisa, Pisa, Italy. [4]Department of Chemistry, Imperial College London MSRH, London, UK. [5]Department of Chemistry, University of Antwerp, Antwerp, Belgium. [6]School of Chemistry, Cardiff University, Cardiff, UK. ✉e-mail: g.d.pantos@bath.ac.uk

dopants to conventional achiral polymers. The latter approach is particularly attractive as chirality can be added to the emissive properties of known achiral polymers without the need for bespoke polymer synthesis[24,25]. Chiral dopants such as aza[6]helicene[26] and binaphthalene-derivatives[21] have been used in this way to great effect, inducing large chiroptical responses in the achiral emissive polymer poly(9,9-dioctyl-fluorene-*alt*-benzothiadiazole) (F8BT)[27].

With scalable access to the chiroptical properties of molecules bearing enantiopure helical or figure-eight topographies clearly desirable, we felt that self-assembly to lemniscular systems may be achievable. Inspection of a figure-eight structure suggests that a twofold disconnection reveals two equidirectional helices, thus, if suitable functionality was present within the helical components, the potential for self-assembly to a figure-eight would be tangible. As such, this report describes our efforts to prepare a suitable enantiopure [5]helicene double electrophile **7** and convert to a figure-eight molecule **8**, demonstrating a simple helicene-based strategy for the synthesis of molecular lemniscates. The sense of helicity will be controlled by a point-to-helical chirality transfer from readily available single-enantiomer propargylic alcohol building blocks, incorporating synthetic scalability and obviating the need for an optical resolution. Having previously applied the alkyne [2 + 2 + 2] cycloisomerisation strategy of Stará and Starý[28,29] to helicene-based pyridines[30,31], this diastereospecific point-to-helical chirality transfer strategy will now be applied to the construction of our functionalised [5]helicenoid scaffold **6**. Subsequently, helicenoids **7**, as well as lemniscate **8**, will be used as chiral dopants in F8BT polymers to produce chiroptically active thin films.

## Results

The diester functionalised [5]helicenoid **6** was synthesised over seven steps via the pathway outlined in Fig. 1. The Sonogashira coupling of **1** and **2** affords the symmetrical diarylalkyne **3**, after which acidic removal of the MOM protecting groups yields bisphenol **4**. This deprotection step proved particularly sensitive and was ultimately successful only with precise control of reaction time. The subsequent double-Mitsunobu reaction of **4** with enantiopure (*R*)-4-phenylbutyn-2-ol forms triyne (*S,S*)-**5**, and a Rh(I)-catalysed cycloisomerisation of (*S,S*)-**5** yields helical diester *P*(*S,S*)-**6** in an excellent yield of 38% over 7 steps. *P*(*S,S*)-**6** was cleanly recovered as a single

diastereomer via column chromatography and unambiguously identified by single-crystal X-ray diffraction (Fig. 2A). The enantiomeric helicene *M*(*R,R*)-**6** was synthesised in an identical synthetic sequence using (*S*)-4-phenylbutyn-2-ol (37% over 7 steps, see Supplementary Information). Subsequent LiAlH₄-mediated ester reduction followed by Swern oxidation enabled the conversion of **6** into the dialdehyde functionalised scaffold **7** in good yield (77% and 74% for *P*(*S,S*)-**7** and *M*(*R,R*)-**7**, respectively) providing a suitable double electrophile for our lemniscate formation strategy.

In line with previously reported literature[29,30], the stereochemical outcome of the cyclisation reaction to form **6** is founded upon the minimisation of 1,3-allylic-like strain between the methyl groups at the stereocentres and the distal phenyl groups. As such, good levels of diastereoselectivity were observed, with a diastereomeric ratio of *ca.* 9:1 determined by [1]H NMR analysis of the crude reaction mixture. No evidence of epimerisation of compound **6** is observed upon long-term storage at room temperature or upon heating to 140 °C for a period of 24 h. Furthermore, computational studies indicate an energy barrier to epimerisation from *P*(*S,S*)- to *M*(*S,S*)-**6** of 33.2 kcal mol⁻¹ (see Supplementary Information), thus demonstrating the conformational stability imparted to the architecture by the dual sp³ stereocentres contained within the two pyran rings of **6**.

The identities of all structures were confirmed using conventional techniques (multinuclear NMR, high-resolution mass spectrometry (HR-MS) and IR spectroscopy), with the molecular structure of *P*(*S,S*)-**6** further confirmed by single crystal X-ray diffraction. Single crystals of *P*(*S,S*)-**6** suitable for X-ray diffraction analysis were grown from a solution of ethyl acetate upon standing at 4 °C. The molecule crystallises in the chiral orthorhombic space group *P*2₁2₁2₁. The helical distortion present in the structure can be defined by the interplanar angle and the average torsion angle. According to the X-ray molecular structure (Fig. 2A), the interplanar angle between the terminal rings is 31.31°. In comparison, the interplanar angle in [5]helicene is 46.0°. The average torsion angle (or twist angle) along the inner rim of the helix, defined as the average of three dihedral angles at the inner rim, was calculated to be 22.63° for helicenoid *P*(*S,S*)-**6**[32]. This is consistent with the literature values reported for [n]helicenes, where a very similar value of 22.29° was reported for [5]helicene[32,33]. As a result of the rigid helical backbone, the ester groups are held in close proximity with just 3.94 Å between the carbonyl carbon atoms.

**Fig. 1 | Synthetic route to lemniscate (*P,P*)−8.** Enantiomeric lemniscate (*M,M*)−**8** was synthesised in an identical synthetic sequence using (*S*)−4-phenylbutyn-2-ol in step 4 (see Supplementary Fig. 1).

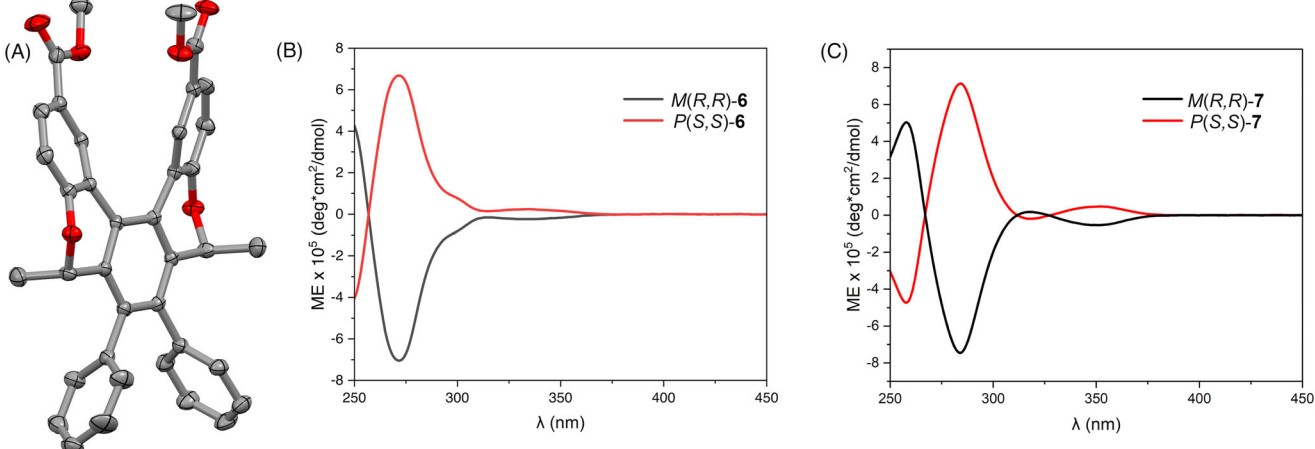

**Fig. 2 | Solid-state structure of 6 and ECD spectra of 6 and 7. A** The molecular structure of [5]helicenoid **6**. The thermal ellipsoids are drawn at 50% probability level and all H atoms are omitted for clarity; (**B**) ECD spectra (molar ellipticity) of compounds *P(S,S)*−**6** (red) and *M(R,R)*−**6** (black) and (**C**) *P(S,S)*−**7** (red) and *M(R,R)*−**7** (black) recorded in CHCl₃.

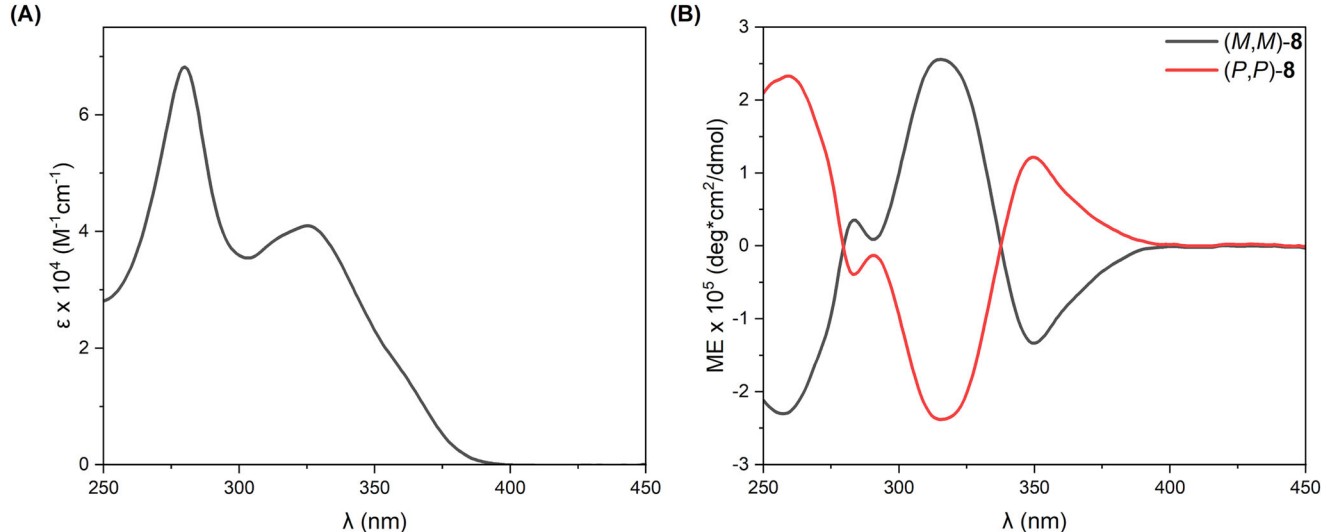

**Fig. 3 | Spectral properties of 8. A** UV-Vis absorption spectrum of (*P,P*)-**8** and (**B**) ECD spectra (molar ellipticity) of (*P,P*)-**8** (red) and (*M,M*)-**8** (black) recorded in CHCl₃.

The ECD spectra of **6** and **7** (Fig. 2B, C, respectively) demonstrate excellent symmetry between the enantiomeric pairs, with a red-shift of *ca*. 10 nm observed in the spectrum of dialdehyde **7** relative to diester **6**.

Having successfully synthesised the electrophilic helicene-derived building block, dialdehyde **7**, the following step was to attempt coupling with a dinucleophilic linker unit. As such, hydrazine was chosen as a reactive and simple nucleophilic diamine linker for initial proof-of-principle studies[34–36]. Accordingly, helicenoid **7** was reacted stoichiometrically with hydrazine monohydrate, leading to the smooth formation of the target azine **8** in high yield (95% (*P,P*)-**8**, 93% (*M,M*)-**8**). Importantly, (*P,P*)-**8** and (*M,M*)-**8** were synthesised using analogous procedures from *P(S,S)*-**7** and *M(R,R)*-**7** respectively (shown in Fig. 1 for the reaction of *P(S,S)*-**7**), enabling independent access to the enantiomeric compounds.

Compound **8** was characterised in full via conventional techniques (multinuclear NMR, HR-MS and IR spectroscopy) to confirm the formation of the macrocyclic [5]helicenoid dimer. A high level of molecular symmetry is reflected in both the ¹H and ¹³C NMR spectra of **8**, whereby a quarter of the number of resonances are observed for the number of protons present in the structure. Disappearance of the aldehyde resonance (observed at 9.62 ppm in the ¹H NMR spectrum of **7**, Supplementary Figs. 24 and 26) confirms complete reaction of the dialdehyde starting material while a singlet resonance at 8.06 ppm is assigned to the four equivalent protons of the newly formed HC = N functionalities (Supplementary Figs. 28 and 30). Given that enantiomers (*P,P*)-**8** and (*M,M*)-**8** were independently synthesised, ECD spectra were obtained for both, with excellent mirror-image replication observed (Fig. 3B). The spectra display three major Cotton effects at 258 nm (|molar ellipticity| = $2.3 \times 10^5$ deg·cm²·dmol⁻¹), 316 nm (|molar ellipticity| = $2.6 \times 10^5$ deg·cm²·dmol⁻¹) and 350 nm (|molar ellipticity| = $1.4 \times 10^5$ deg·cm²·dmol⁻¹) with an additional small band observed at 283 nm.

The synthesis of **8** is based on imine formation, a reaction commonly used in dynamic combinatorial chemistry. As such, this allowed us to study the lemniscate formation in a dynamic combinatorial library, in which 1 equiv of *rac*-**7** was mixed with 1 equiv of hydrazine. The results of HPLC analysis of the DCL (isocratic CH₃CN, ChiralPak IB, see Supplementary Information) show a self-sorting behaviour, resulting in formation of the two enantiomeric lemniscates, while the formation of the heterochiral macrocycle containing both *P(S,S)*- and *M(R,R)*-**7** was not observed.

Further support for the structural assignment of **8** came from molecular modelling. Accordingly, the first requirement was to perform an energy minimisation on **8**, with initial findings showing a strong preference for **8** to adopt a $D_2$-symmetric conformation, analogous to many of the lemniscular structures reported in literature[6,9,12]. An E/Z and atropisomeric search of possible conformers confirmed the $D_2$-symmetric figure-eight structure (Fig. 4A) to be the lowest free energy conformation of **8**.

Subsequently, UV-Vis, ECD, IR and VCD profiles were simulated for the $D_2$-symmetric conformation of **8** and compared to experimental data (Fig. 4). Simulations of the spectra were performed using sTDDFT methods (see Supplementary Information for details). The simulated UV-Vis and ECD spectra (Fig. 4B, C) show excellent fit to the experimental data which is further corroborated in the IR and VCD data (Fig. 4D). Having synthesised enantiopure samples of both helicenoid and lemniscate derivatives, we next sought to investigate their application in chiroptical materials. Enantiomeric helicenoid dialdehydes M(R,R)-**7** and P(S,S)-**7** and lemniscate (M,M)-**8** and (P,P)-**8** were used as chiral dopants in thin films of the achiral polymer poly(9,9-dioctylfluorene-alt-benzothiadiazole) (F8BT). Thin films were prepared from solutions of F8BT and dopants in toluene (at 35 mg/mL, 9:1 F8BT:[5]helicenoid w/w ratio, which, as the MWt of 7 is half of that of 8, equates to the same amount of "helical content" for both 7 and 8). The solutions were spin-coated on square fused-silica substrates and the films annealed for 60 min at 170 °C. The chiral response of the films was measured before and after annealing, confirming complete

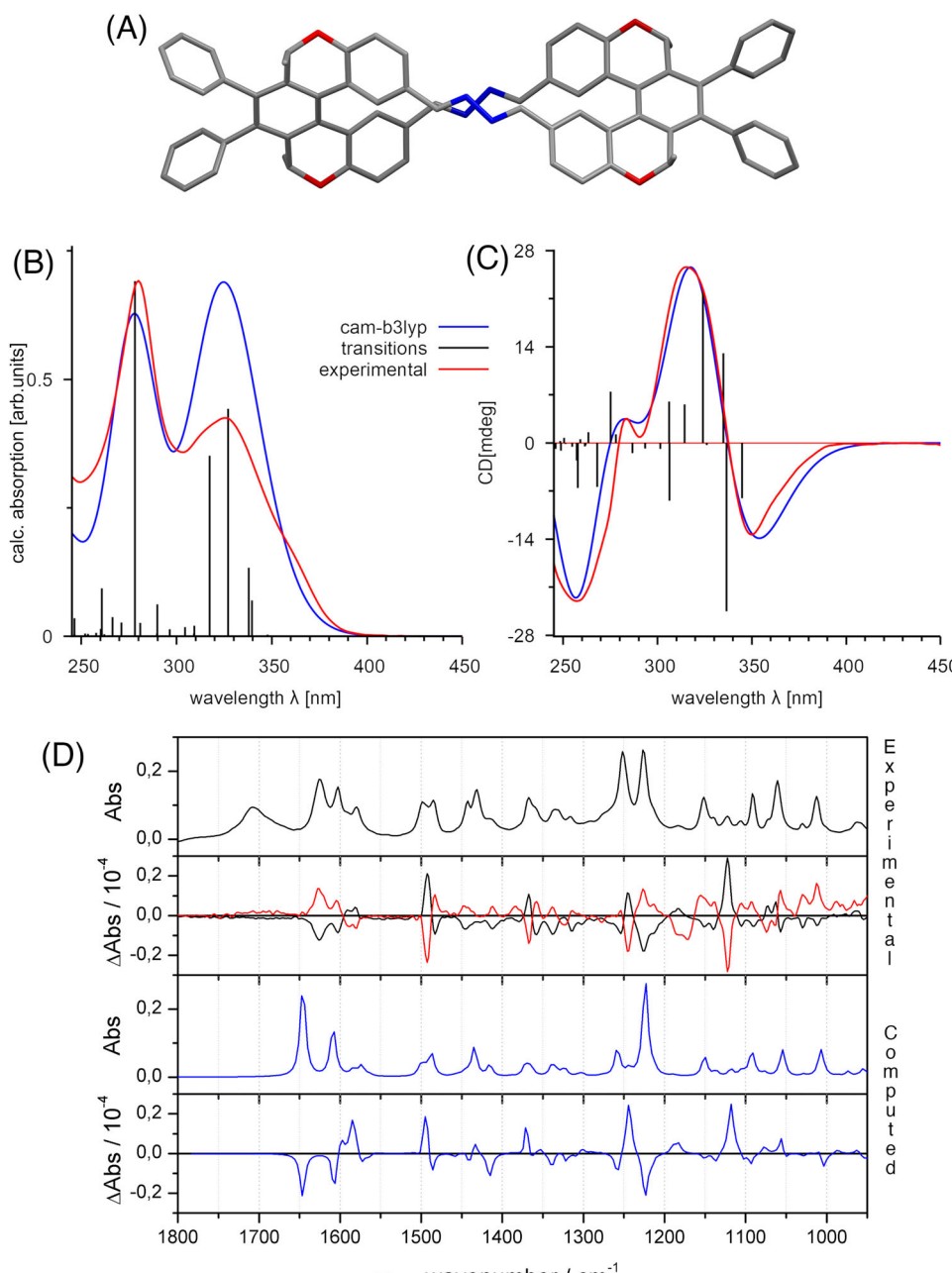

**Fig. 4 | Structure of (M,M)-8 and experimental and computed spectra of 8.**
**A** $D_2$-symmetric energy minimum model of (M,M)-**8** with H atoms omitted for clarity. **B** UV-Vis and (**C**) ECD of (M,M)-**8** showing comparison of the experimental (CHCl₃, red) and simulated (cam-B3LYP/D3BJ/def2-TZVP/def2-J/SMD(CHCl₃) blue) spectra. **D** Comparison of the experimental IR (solid) and VCD spectra (CDCl₃, (P,P)-**8** black, (M,M)-**8** red) with the simulated spectra (B3LYP/D3BJ/def2-TZVP/CPCM(CHCl₃) blue) of (P,P)-**8**.

annealing due to the lack of free [5]helicenoid transitions in the CD spectra after thermal treatment. Large Cotton effects (>3000 mdeg) were observed on the F8BT transitions in the thin films containing [5]helicenoids M(R,R)- and P(S,S)-7, respectively (Fig. 5). Pleasingly, opposite Cotton effects were observed in films doped with enantiomeric helicenoids, indicating transfer of chirality from the chiral dopant to the thin film. The larger Cotton effect is translated to large dissymmetry factors for the helicenoid/F8BT films: $g_{abs}$ = ±0.169 at 500 nm for 7/F8BT (Fig. 6B).

The thin films containing lemniscate **8** and F8BT displayed even larger Cotton effects post-annealing, with recorded ellipticity of 7000 mdeg at 489 nm (Fig. 6A) and a dissymmetry factor, $g_{abs}$, of 0.287 for (P,P)-**8** and -0.358 for (M,M)-**8**, both recorded at 492 nm (Fig. 6B). This increase in $g_{abs}$ compared to **7**/F8BT films demonstrates the benefit of the lemniscular shape in inducing dissymmetry.

The doped F8BT thin films were also analysed with the aid of Mueller Matrix Polarimetry (MMP)[37–39] carried out at Diamond B23 beamline. MMP enables assessment of the linear dichroism and birefringence (LD and LB) contributions to the response observed on the conventional, bench-top CD spectropolarimeters. MMP analysis of our

films shows little-to-no LD and/or LB contributions, suggesting that the observed signal is a true CD with no artefacts (see Supplementary Information). This confirms that the helicenoids induce a certain handedness onto the polymer depending on the helical chirality introduced via synthesis. The films also show no sign of depolarisation; the Mueller elements M03 and M30 (corresponding to the CD) show the same sign, whereas M12 and M21 (corresponding to the CB) display opposite signals. All these factors suggest a relatively straightforward optical behaviour, indicating that the method chosen to analyse the data gives chiroptical elements free from artefacts. We have also recorded a $1 \times 1\,mm$ MMP grid scan of our films (mapping of $21 \times 21$ points) taking advantage of the highly collimated beamlight (50 μm in diameter) available at B23 beamline at Diamond. The map showed a uniform and homogeneous chiral response across the mapped area of the film, indicating that the films are uniform in terms of chiroptical response.

Given that the generation of circularly polarised luminescence is a desirable property for chiroptical materials, we sought to utilise our systems to produce thin films that exhibit such a response. While F8BT polymer is emissive, unsurprisingly, the thin film containing only the achiral F8BT exhibited no CPL. However, pleasingly, thin films doped with both enantiomers of helicenoid **7** and lemniscate **8** demonstrate substantial CPL responses (Fig. 7). The CPL response was observed on the F8BT emission rather than on the chiral dopant emission as the latter is obscured by the F8BT broad absorbance between 300 to 500 nm and is characterised by low $g_{lum}$ measured in CHCl$_3$ solution (see Supplementary Information). The quantum yield of the F8BT thin films[40] increased from 9% to 11% and 15% upon doping with **7** and **8**, respectively. The excited state analysis (CPL) of the films matches that performed in the ground state (CD), with the film doped with lemniscates displaying the largest dissymmetry in the excited state (Fig. 7). The $g_{lum}$ for the films doped with P(S,S)- and M(R,R)-**7** were measured at +0.151 and −0.204 (511 nm, respectively), while the $g_{lum}$ recorded for the films doped with (P,P)- and (M,M)-**8**, was +0.267 and −0.346 at 511 nm, respectively. These films are stable to temperature variation as by the little variation in the $g_{lum}$ values over a 105 °C temperature range (see Supplementary Information).

## Discussion
The production of emissive chiral thin films is predicated by the efficient synthesis of helical and leminscular chiral dopants. The synthetic protocol described here allows the enatiopure synthesis of a [5]helicenoid scaffold in 38% yield over 7 steps from commercially available

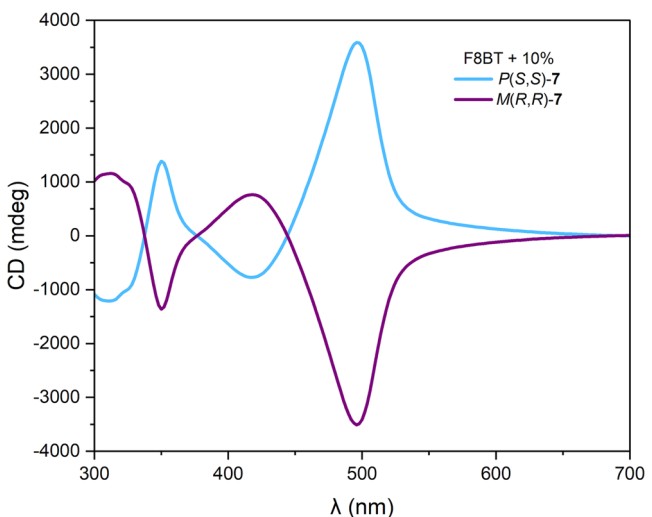

**Fig. 5 | ECD spectra of thin films of F8BT with 10% 7.** Light blue: P(S,S)−**7** and purple: M(R,R)−**7**, respectively.

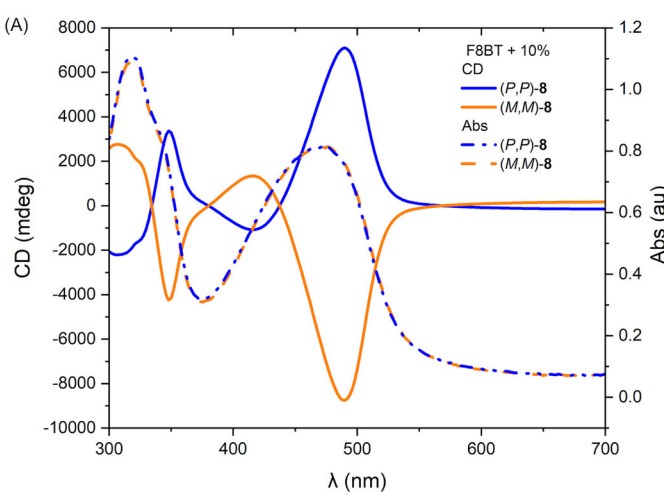

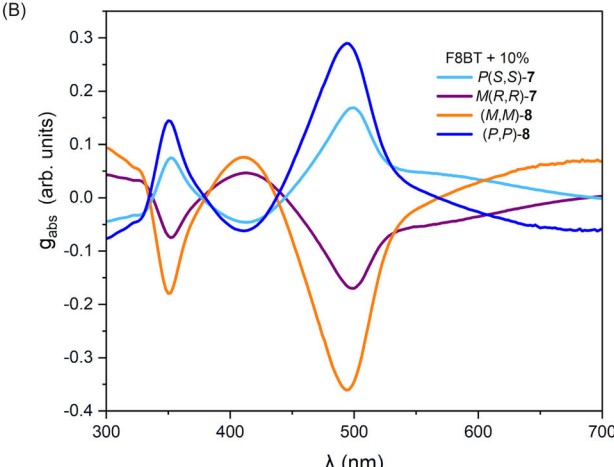

**Fig. 6 | Absorption, ECD spectra and dissymmetry factors of thin films of F8BT with 10% chiral dopant. A** ECD spectrum (solid trace) and absorption spectrum (dashed trace) of F8BT with 10% of (P,P)-**8** (blue) and (M,M)-**8** (orange), respectively and (**B**) $g_{abs}$ spectra of F8BT thin films with 10% (P,P)-**8** (blue), (M,M)-**8** (orange), P(S,S)−**7** (light blue), M(R,R)−**7** (purple), respectively.

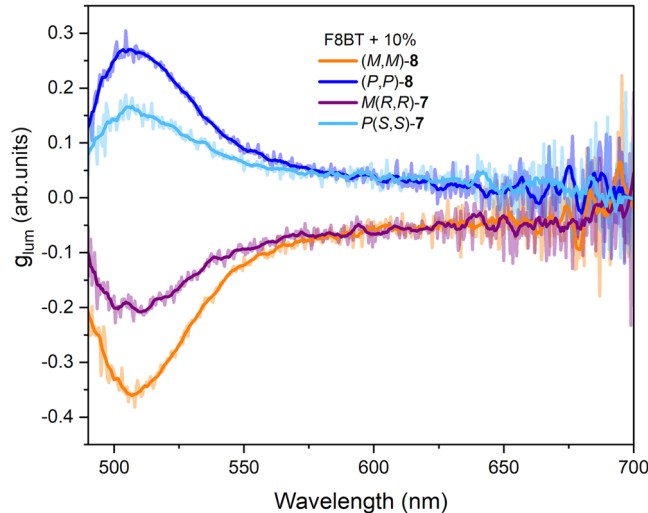

**Fig. 7 | CPL spectra of thin films of F8BT with 10% chiral dopant.** $g_{lum}$ spectra of F8BT thin films with 10% of $(P,P)$−**8** (blue), $(M,M)$−**8** (orange), $P(S,S)$−**7** (light blue), $M(R,R)$−**7** (purple), respectively. The faded lines represent the raw data and non-transparent lines represent smoothed data. See Supplementary Information for the ΔI data.

materials. When incorporated into F8BT films in a ratio 1:9 helicenoid:F8BT, the enantiomeric helicenoids **7** lead to the formation of chiral thin films with large dissymmetry factors in both the ground ($g_{abs}$ = ±0.169 for **7**) and excited state ($g_{lum}$ = +0.151 or −0.204 for $P(S,S)$- or $M(R,R)$-**7**, respectively) making them attractive CPL active materials.

Further reaction of the dielectrophilic [5]helicenoid **7** with hydrazine afforded the lemniscular bisazine **8**, the molecular structure of which was investigated via conventional chemical characterisation techniques and extensive circular dichroism studies (electronic and vibrational). By taking advantage of the asymmetric helicene synthesis described, we negate the need for optical resolution at any point within the synthesis of **8**, thus demonstrating an enantiopure route to helicene-derived molecular lemniscates.

The figure-eight structure of this molecule imparts chirality in F8BT films that have even larger dissymmetry factors than the films produced from the constitutive helicenoids while having the same amount of 'helical content', with $g_{abs}$ = 0.287 and −0.358 recorded at 492 nm for $(P,P)$- and $(M,M)$-**8**, respectively. The doped films displayed remarkable CPL activity with a $g_{lum}$ = 0.267 and −0.346 recorded at 511 nm for $(P,P)$- and $(M,M)$-**8**, respectively, and high thermal stability, making them ideally suited for studies in chiral photonic applications. The modularity of the synthetic approach described enables the exploration of both alternative linkages and helical scaffolds. Given the promising initial chiroptical results, this, in principle, would provide an easy-access route to a diverse array of helicene-derived lemniscates likely to be of interest as chiroptical materials.

## Methods

### Synthetic considerations
All reagents were purchased from the commercial suppliers Acros Organics, Alfa Aesar, Sigma Aldrich, TCI Europe, Fluorochem and Ossila and used without further purification.

Reactions involving the use of moisture-sensitive reagents were carried out under an inert atmosphere ($N_2$ or Ar) using anhydrous solvents. Solvents were dried on an Innovative Technology PS-400-7 Solvent Purification System.

Thin layer chromatography (TLC) was performed using commercially available silica plates (pre-coated, aluminium-backed). Visualisation was accomplished by direct observation under UV light (254 nm). Flash chromatography was performed using a Büchi

RevelerisX2® flash system, with prepacked GraceResolv® silica cartridges and the solvent system as stated. Where manual flash column chromatography was conducted, the solid phase is assumed to be Sigma Aldrich Silica gel of pore size 60 Å.

### Film preparation
Thin films were prepared using a Laurell Model WS-650HZ-23NPP/ LITE spin coater. The films were produced using F8BT and the chiral dopant in a 9:1 mass ratio using toluene (35 mg/mL). In earlier experiments the amount of chiral dopant was varied between 9:1, 8:2 and 7:3 with the 9:1 ratio giving the best chiral induction, this is in line with the behaviour of other F8BT/chiral dopant mixtures[41]. The films were spin-coated using 70 µL. The spin rates were as follows: 12,000 RPM and 2000 acceleration RPM/s. After preparation, the films were annealed for 1 h at 170 °C. The solutions for films with $(M,M)$-**8** and $(P,P)$-**8** were heated to 100 °C to solubilise the cloudy solution (($M,M$)-**8** and $(P,P)$-**8** were not soluble at that concentration at that temperature). Each film was prepared in triplicate.

The F8BT/chiral inducer blends contain the same amount of helical unit in each case. The inducers (either **7** or **8**) represent 10% of mass in the mixture. As the mixture concentration is 35 mg/mL, the inducer concentration is 3.5 mg/mL. Converting the mass value to molarity, we obtain the following concentrations: 6.7 µmoles/mL for inducer **7** and 3.3 µmoles/mL for inducer **8**. Because the inducer **7** has only one helix, the molarity value for **7** can be considered as molarity of the helical unit. On the other hand, inducer **8** contains two helices thus it gives a molarity of helical unit of 6.6 µmoles/mL. In summary, the spectra in the manuscript for F8BT blended with inducers **7** and **8** are represented as 'per helical unit'. With the same number of helical units, the CD and CPL spectra of F8BT blended with 10% of enantiomeric inducer **8** showed overall a higher CD intensity magnitude than those blended with inducer **7** which highlights the benefit of doping with a lemniscular structure.

### Nuclear magnetic resonance
$^1$H (all 1D and 2D experiments) and $^{13}$C NMR spectra were acquired on Bruker Avance 300 MHz ($^1$H 300 MHz and $^{13}$C 75 MHz), Bruker Neo 400 MHz ($^1$H 400 MHz and $^{13}$C 101 MHz), Bruker Avance 400 MHz ($^1$H 400 MHz and $^{13}$C 101 MHz) or Agilent ProPulse 500 MHz ($^1$H 500 MHz and $^{13}$C 125 MHz) systems as stated. Chemical shifts (δ) are reported in parts per million (ppm) relative to tetramethylsilane (TMS; δ = 0.00). All spectra were referenced to the residual solvent peaks and acquired at 298 K or at the temperature specified in each case. Samples for NMR spectroscopy were prepared with acetone-$d_6$, chloroform-$d$ (CDCl$_3$), dimethylsulphoxide-$d_6$ (DMSO-$d_6$), methanol-$d_4$ (CD$_3$OD) and 1,1,2,2-tetrachloroethane-$d_2$ (TCE-$d_2$) as stated in each case. Coupling constants (J) are reported in Hertz (Hz) and signal multiplicities are denoted using standard nomenclature as broad singlet (br s), singlet (s), doublet (d), doublet of doublets (dd), doublet of triplets (dt), triplet (t), triplet of triplets (tt), quartet (q), pentet (p), multiplet (m), etc.

### High-resolution mass spectrometry
Electrospray ionisation quadrupole time-of-flight (ESI-Q-TOF) mass spectrometry was performed on an Agilent Technologies 6545 Q-TOF LC-MS instrument using a positive- or negative-ion mode as stated in each case.

### FT-IR spectroscopy
Infra-red (IR) spectroscopy was carried out using a Perkin Elmer Spectrum 100 FT-IR system (16 scans, 1 cm$^{-1}$ resolution). Solid samples were prepared directly on the ATR plate either neat or as a film.

### Melting point determination
Melting points were obtained on a Stuart SMP-10 capillary melting point apparatus or an Optimelt automated melting-point system (Stanford Research Systems).

## Single crystal X-ray diffraction

Single crystal X-ray diffraction data were collected on a Rigaku Gemini A Ultra dual-source diffractometer, equipped with an Atlas CCD detector and an Oxford Instruments Cryojet-XL for temperature control, using $MoK_{\alpha}$ radiation at 100 K. Data collection, indexing and integration procedures were performed using Rigaku control software CrysAlisPRO. Structure solution was performed by dual-space algorithm using the programme SHELXT[42] and refined by full-matrix least-squares against $F^2$ using SHELXL[42], both implemented through the Olex2 interface (v1.5). Hydrogen atoms were placed geometrically and refined using a riding model, with $U_{iso} = 1.2$ x the parent atom for CH and $U_{iso} = 1.5$ x the parent atom for $CH_3$.

## High-performance liquid chromatography

HPLC analysis was performed on an Agilent Technologies 1100 Series system coupled to an Agilent 1260 Infinity Diode Array Detector and a JASCO OR-1590 chiral detector. Chromatographic separation of a 5 µL sample injection was performed on a Daicel CHIRALPAK IB 5 µm, 4.6 × 250 mm column using 100% acetonitrile (VWR, HiPerSolv) as the mobile phase. The column was operated at a flow rate of 1 mL min$^{-1}$ at 35 °C with a run time of 30 min.

## Electronic circular dichroism spectroscopy

Absorption and electronic circular dichroism (ECD) experiments were performed on an Applied Photophysics Chirascan Circular Dichroism spectrophotometer equipped with a Peltier temperature controller using quartz cuvettes (Starna Scientific Ltd) of 1 or 10 mm pathlength in $CHCl_3$. The background corresponding to the solvent and cuvette's absorption was recorded and subtracted from subsequent measurements. Wavelengths are reported in nanometres. The experiments were conducted using a 1 nm step acquisition, with time per point of 0.5 s and bandwidth 1 nm.

## Vibrational circular dichroism spectroscopy

Fourier transform infra red (FT-IR) and vibrational circular dichroism (VCD) spectroscopy were collected simultaneously using a Biotools® ChirallR-2X (Biotools Inc., Jupiter, FL) with a liquid nitrogen cooled MCT detector. Samples were prepared by dissolving each enantiomer in deuterated chloroform ($CDCl_3$) at a concentration of 28 mM and subsequently transferring the solution to a fixed path length (100 µm) measuring cell (fitted with calcium fluoride windows). The spectra were collected at 4 cm$^{-1}$ resolution and 20,000 scans.

## Circularly polarised luminescence

CPL spectra were acquired on a benchtop B23 Olis Hummingbird-based CPL spectrophotometer or an Applied Photophysics Chirascan Circular Dichroism spectrophotometer equipped with a CPL accessory and a Peltier temperature controller. Both instruments are working in 90° configuration. For the experiments conducted on B23 Olis-CPL, the excitation wavelength of unpolarised light at 460 nm. Wavelength range was 480–700 nm, every 0.5 nm, 10 s integration time and 1 repetition. The excitation slit width was 5 mm and the emission slit width was 3 mm. The film was facing the CPL Photoelastic modulator (PEM) with the excitation wavelength irradiating the thickness of the fused silica substrate on which the film was spin-coated. For the experiments conducted on Chirascan CPL as a function of temperature (Supplementary Figs. 67 and 68), the excitation wavelength was 460 nm using light linearly polarized in the horizontal plane. Wavelength range was 480–700 nm, every 1 nm, 10 s integration time. The excitation bandwidth was 24 nm, and the emission slit width was 14 nm. Like for the B23 Olis-CPL, the film was facing the CPL PEM.

## Mueller matrix polarimetry (MMP)

The MMP experiments were carried out at Beamline B23, Diamond Light Source on a custom build Mueller Matrix Polarimeter (Hinds Instruments, Hillsboro, OR, USA) attached to a double grating subtractive monochromator (Olis, Athens, USA)) using the highly collimated beamlight generated at B23 beamline module station B as light source[38]. The MMP data was acquired in transmission mode, from 650 to 300 nm, every 2 nm. The maps (1 × 1 mm) were acquired at 50 µm resolution, which was possible due to the highly collimated beam at Beamline B23. The MMP instrument has the option to run in either MMP or CD mode. All CD mode experiments were run using the same parameters and the same area of MMP mode for a comprehensive comparison.

## Quantum yield measurement

The quantum yields were determined using a Horiba Fluorolog-QM spectrofluorometer equipped with QuantaPhi-2 integrating sphere. The quantum yields for the films were recorded with the film positioned in a customised powder holder with the following parameters: excitation wavelength 460 nm, emission scan 480–800 nm, every 1 nm, excitation and emission bandwidths of 3 nm, and a time per point of 0.1 and 1 s, respectively. The quantum yields for the F8BT solutions were recorded under the same conditions except for using the QuantaPhi-2 cuvette holder. The quantum yield for films of F8BT with dopants is reported as the average of 12 measurements (2 times per point for each film, three films for each enantiomer of the dopant).

## Data availability

The data that support the findings of this study are available in the paper and its Supplementary Information. The experimental and computational datasets generated during this study are available at Imperial College Research Data Repository (https://doi.org/10.14469/hpc/3901)[43]. CCDC 2328087 (**6**) contains the supplementary crystallographic data for this paper. This data can be obtained free of charge from The Cambridge Crystallographic Data Centre via www.ccdc.cam.ac.uk/data_request/cif. All data are available from the corresponding author upon request.

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

## Acknowledgements

We thank the Engineering and Physical Sciences Research Council (EPSRC) DTA for L.E.M.W, EPSRC Programme Grant (EP/K004956/1), University of Bath Major Equipment Fund (VB-FS1RCU) and Diamond Light Source for the beam time and instruments access to B23 beamline (CM33876-1).

## Author contributions

Conceptualisation, D.R.C. and G.D.P.; Methodology, G.D.P., D.R.C. and G.S.; Investigation, L.E.M.W., T.M.G., F.P.C., C.F., A.T., H.S.R., C.J., L.E.H. and G.D.P.; Writing—Original Draft, L.E.M.W., T.M.G., D.R.C. and G.D.P.; Writing—Review & Editing, F.P.C., C.F., H.S.R., C.J., L.E.H. and G.S.; Funding Acquisition, D.R.C., G.S. and G.D.P.; Resources, H.S.R., C.J., G.S. and G.D.P.; Supervision, G.S., D.R.C., and G.D.P.

## Competing interests

The authors declare no competing interests.
