## [Transparent Peer Review file · Nature Communications]

Enantiopure Synthesis of [5]Helicene Based Molecular Lemniscates and Their Use in Chiroptical Materials

Corresponding Author: Dr G. Pantos

Version 0:

Reviewer comments:

Reviewer #1

(Remarks to the Author)

Pantos and coworkers prepared the manuscript entitled "Enantiopure Synthesis of [5]Helicene Based Molecular Lemniscates and Their Use in Chiroptical Materials". The authors successfully synthesized enantiomeric heliceneoids 6 by using chiral 4-phenylbutyn-2-ol, and the chiral derivative 7 was condensed with hydrazine to furnish the figure-eight dimeric heliceneoid 8 in excellent yield.

Although 8 themselves were CPL inactive, they were found to function as good chiral dopants for the film of F8BT.

Considering the elegant synthesis and good chiroptical properties of the film, this manuscript might be published in Nature Communications. Nevertheless, the following points should be considered.

- 1) X-ray structure(s) of 8 is needed for publication.
- 2) For CPL, both plots of ΔI vs wavelength and g_{lum} vs wavelength should be included better understanding for readers.
- 3) FL spectra and quantum yields of 7, 8, and films (F8BT with/without 7 or 8) should be included because they will help the discussion.
- 4) The following papers should be cited.
Asymmetric systematic synthesis of dihetero[8]helicenes (Chem. Sci., 2021, 12, 2784)
Synthesis of cyclic azahelicene dimers (Angew. Chem. Int. Ed. 2024, 63, e202404149)

Reviewer #2

(Remarks to the Author)

The manuscript by Pantos and co-workers reports the enantiopure synthesis of [5]helicenes and the dimerized molecular lemniscates, and corresponding chiroptical characterization. The first interesting point here is to control the helical chirality by the point chirality of building blocks used in the [2+2+2] triyne cycloisomerization. The single crystal x-ray structure of helicene 6 is reported that further confirms the enantioselectivity of cycloaddition reactions. After the elegant synthesis of the [5]helicene and helicene-derived figure-eight dimers, the chiroptical properties are then discussed with high numbers of g_{lum} values.

The first question goes to the synthetic aspects. As expected for this group of [5]helicene, the design and synthesis are very well described and elegant. This study is built on the previous work (as cited in ref 27, 28 and other in the literature), and a main contribution is the enantiopure lemniscate. While it is very nice, there is not a breakthrough in synthetic chemistry.

Regarding the aspects of chiroptical properties in the ground and excited states, the authors claimed large dissymmetry factors up to 0.358 for the thin films. However, I have a concern that these values may not be reliable as many conditions play important roles in the chiroptical measurements of solid-state materials. These values are reproducible? The authors require to acquire chiroptical properties in solutions as well, and it is crucial to clearly explain why these molecules show remarkable g values by computations.

Therefore, from the view of this reviewer, the manuscript may not be significant enough for publication in top journals

including nature communications.

Reviewer #3

(Remarks to the Author)

The manuscript from Pantoş and co-workers describes the first synthesis of enantiopure lemniscates and their chiroptical properties including their activity as polymer dopants. Synthetically, the work relies on the methodology developed by the Stary/Stara group, and the macrocyclization approach is not new either (some work on hydrazine linked macrocycles could be cited). However, the true relevance of this work is in the CPL results: the lemniscates turn out to be much better dopants than the open macrocycles. I believe the authors could discuss the quantitative aspect of doping in a bit more detail: they use w/w doping ratios which incidentally lead to comparable amounts of helical units in all blends (i.e. there is one helix per molecule in 6 and 7, and two helices in 8). However explicit discussion of the "helical content" in the blends would help appreciate the relevance of the result.

This is also one of the best-edited manuscripts I reviewed in a long while. The writing is clear and succinct, and the text and graphics are essentially free from errors. Overall, I recommend publication after a minor revision.

Version 1:

Reviewer comments:

Reviewer #1

(Remarks to the Author)

The authors revised the manuscript according to the reviewer's comments to some extent, and the present manuscript has been improved. This manuscript may be accepted in Nature Communications in future. But the reviewer still has comments below.

a) As the authors state, the absolute configuration of 8 has already determined by the VCD spectrum of 8 and the crystal structure of 6.

The reviewer suggested the single X-ray diffraction analysis of 8 in order to clarify the solid-state structure rather than to determine the absolute configuration. Did the authors try the crystallization of the mixture of (P,P)-8 and (M,M)-8? General helicenes crystallize in the racemic form, in which (P)-helicene and (M)-helicene are often stacked in the crystal packing. Generally, racemic molecules are easy to crystalize as compared to the enantiomeric forms.

b) Keywords in page 1, "Circularly Polarised Luminescence" is better.

Reviewer #2

(Remarks to the Author)

Thanks for the authors' responses to all the reviewers. I appreciate the author have addressed some of my concerns. This reviewer fully agree that molecular modelling of F8BT doped thin films is challenging. On my opinion, however, the experimental glum values should be acquired for molecules 7 and 8 if they are luminescent. And relevant computations should be performed to disclose the mechanism of remarkable g values. I would like the authors to consider this comment.

Version 2:

Reviewer comments:

Reviewer #1

(Remarks to the Author)

The authors responded to the almost all the reviewer's comments, and the present manuscript has been improved. In my opinion, this manuscript can be accepted in Nature Communications in the present form.

Reviewer #2

(Remarks to the Author)

The authors' responses are appreciated. However, on the opinion of this reviewer, the computational g factors are possible for small molecules (7 and 8), and also should be acquired regardless of the experimental glum values. It is exciting that the authors "would like to contribute to the understanding of this chirality induction and we are currently working towards this". Honestly, this reviewer believes a clear elucidation of mechanisms for high glum values would be one of the critical contributions that make it possible to be published in top journals. At this point, I disagree with the authors' statement "Such an investigation represents a standalone study which, if included in the current manuscript, would take away the focus of this chemistry describing the enantiopure lemniscate synthesis". In the case of only giving an example of chiral thin films with large glum values but without teaching people why you could do that (I am sure our readers are quite interested in this), I would suggest the publication in a more specific journal.

Reviewer #1:

“Pantoş and coworkers prepared the manuscript entitled "Enantiopure Synthesis of [5]Helicene Based Molecular Lemniscates and Their Use in Chiroptical Materials". The authors successfully synthesized enantiomeric helicoids **6** by using chiral 4-phenylbutyn-2-ol, and the chiral derivative **7** was condensed with hydrazine to furnish the figure-eight dimeric helicoid **8** in excellent yield. Although **8** themselves were CPL inactive, they were found to function as good chiral dopants for the film of F8BT. Considering the elegant synthesis and good chiroptical properties of the film, this manuscript might be published in Nature Communications. Nevertheless, the following points should be considered.”

We thank the referee for their positive comments about our work.

“1) X-ray structure(s) of **8** is needed for publication.”

We worked hard before submitting the manuscript to grow x-ray diffraction suitable crystals of the lemniscates. Unfortunately, we were unsuccessful despite year-long attempts. Upon receiving the reviews and saw the request of this referee, we have renewed our efforts to grow crystals of **8**. We have set up crystallisations in Bath, at Diamond Light Source and in Cardiff, by three different co-authors but unfortunately, we have not been successful in producing good x-ray diffraction quality crystals. Lemniscate **8** is only sparingly soluble in dichloromethane and chloroform, and we have explored crystallizations via a variety of methods including evaporative, diffusion and cooling processes. Cooling crystallizations from dichloromethane showed the most promise, however, still produce only extremely small and thin needles that do not show sufficient diffraction even on a state-of-the-art rotating anode CuK α source. However, as we have good VCD experimental and predicted data, that is backing our NMR analysis, we would argue that this is sufficient to determine the structure of **8** in solution, thus rendering the solid-state structure unnecessary. We would like to highlight an article by one of our co-authors showing that correct absolute configuration can be determined from VCD or ROA alone, when in combination with predictions (“A combined Raman optical activity and vibrational circular dichroism study on artemisinin-type products”, DOI: 10.1039/D0CP03257C)

“2) For CPL, both plots of ΔI vs wavelength and g_{lum} vs wavelength should be included better understanding for readers.”

We have included the requested ΔI vs wavelength plots in the ESI (Figure S64) and made a reference in the g_{lum} caption in the main text (Figure 6).

“3) FL spectra and quantum yields of **7**, **8**, and films (F8BT with/without **7** or **8**) should be included because they will help the discussion.”

We have included the calculated quantum yields for the films in the main text and have included a table (Table S2) and the corresponding figures (Figure S71 and Figure S72) in the ESI. The quantum yields for **7** and **8** in CH₃CN solution were below the detection limit of the QuantaPhi-2 integrating sphere from Horiba and therefore could not be determined.

“4)The following papers should be cited. Asymmetric systematic synthesis of dihetero[8]helicenes (Chem. Sci., 2021, 12, 2784) Synthesis of cyclic azahelicene dimers (Angew. Chem. Int. Ed. 2024, 63, e202404149)”

We have included the requested references in the revised version of the manuscript (references 18 and 19).

Reviewer #2:

“The manuscript by Pantos and co-workers reports the enantiopure synthesis of [5]helicenes and the dimerized molecular lemniscates, and corresponding chiroptical characterization. The first interesting point here is to control the helical chirality by the point chirality of building blocks used in the [2+2+2] triyne cycloisomerization. The single crystal x-ray structure of helicene **6** is reported that further confirms the enantioselectivity of cycloaddition reactions. After the elegant synthesis of the [5]helicene and helicene-derived figure-eight dimers, the chiroptical properties are then discussed with high numbers of g_{lum} values.

The first question goes to the synthetic aspects. As expected for this group of [5]helicene, the design and synthesis are very well described and elegant. This study is built on the previous work (as cited in ref 27, 28 and other in the literature), and a main contribution is the enantiopure lemniscate. While it is very nice, there is not a breakthrough in synthetic chemistry.”

We thank the referee for appreciating our elegant synthesis. We believe it is a strength of this work.

“Regarding the aspects of chiroptical properties in the ground and excited states, the authors claimed large dissymmetry factors up to 0.358 for the thin films. However, I have a concern that these values may not be reliable as many conditions play important roles in the chiroptical measurements of solid-state materials. These values are reproducible?”

While we understand the reviewer's concerns, we can confirm that the measured g-lum values are reproducible. We have manufactured three different films for each enantiomer of **7** and **8** and have measured their properties, including the g-lum. We have included the corresponding graphs in the ESI (Figure S63, Figure S67); these data highlight the reproducibility of our film production method.

"The authors require to acquire chiroptical properties in solutions as well, and it is crucial to clearly explain why these molecules show remarkable g values by computations. Therefore, from the view of this reviewer, the manuscript may not be significant enough for publication in top journals including nature communications."

We have included the toluene solution data for F8BT, F8BT+**7** and F8BT+**8** in the ESI (Figure S69). As expected, there is no chirality induction on the F8BT specific electronic transitions (centred at 450 nm), likely due to the fact that both chiral dopants and F8BT are better solvated/have a stronger interaction with the solvent (toluene) which prevents an interaction between the chiral dopant and F8BT in solution.

Regarding the potential molecular modelling study to explain/predict the high g-lum values observed, we agree with the referee that this would be indeed very interesting and useful, however we, and the community, consider this as a separate project. We point the referee to an article published in ACS Nano 2022, 16, 14432 "Conformational Heterogeneity and Interchain Percolation Revealed in an Amorphous Conjugated Polymer" (DOI: 10.1021/acsnano.2c04794) in which the authors describe the very complex problem of predicting the F8BT structure. On top of this complex problem, we would need to include the geometrical influence of the chiral dopant on the tangled structure of F8BT which, in itself, increases the complexity exponentially. Predicting the g-lum values would require extensive parametrisation for predicting the electronic transitions using a semi-empirical method (eg. ZINDO-RPA) which could potentially handle the large number of atoms in the putative structure. Higher level of electronic structure prediction (sTDDFT or TDDFT) would require significantly more computational resources, unlikely to be met by the current technology. Two of the co-authors, Prof. Rzepa and Dr Pantos, have extensive experience (50 years and 25 years, respectively) in molecular modelling and have concluded that this type of computational work would require extensive resources which makes it a separate study that we hope to undertake should the opportunity present itself.

Reviewer #3:

"The manuscript from Pantoş and co-workers describes the first synthesis of enantiopure lemniscates and their chiroptical properties including their activity as polymer dopants. Synthetically, the work relies on the methodology developed by the Stary/Stara group, and the macrocyclization approach is not new either (some work on hydrazine linked macrocycles could be cited). However, the true relevance of this

work is in the CPL results: the lemniscates turn out to be much better dopants than the open macrocycles. I believe the authors could discuss the quantitative aspect of doping in a bit more detail: they use w/w doping ratios which incidentally lead to comparable amounts of helical units in all blends (i.e. there is one helix per molecule in 6 and 7, and two helices in 8). However explicit discussion of the "helical content" in the blends would help appreciate the relevance of the result."

We thank the referee for their positive comments. We have added references to some hydrazine linked macrocycles as suggested (references 34 – 36). Regarding the "helical content", we have added an explanation in the main text, stating that due to the difference in MWt between 7 and 8, and as the referee correctly pointed out, and because our blends are w/w ratios, the amount of "helical content" is quasi-constant in all the films that we have produced. This therefore highlights the fact the lemniscates are much better dopants than the helicenoids. A more extensive explanation has also been included in the ESI (page 3).

This is also one of the best-edited manuscripts I reviewed in a long while. The writing is clear and succinct, and the text and graphics are essentially free from errors. Overall, I recommend publication after a minor revision.

We would like to thank the referee for these comments. They will push us to always produce manuscripts of high quality, a difficult task, but worth it.

We would like to thank the reviewers for their work on our manuscript.

Reviewer #1 (Remarks to the Author):

The authors revised the manuscript according to the reviewer's comments to some extent, and the present manuscript has been improved. This manuscript may be accepted in Nature Communications in future. But the reviewer still has comments below.

a) As the authors state, the absolute configuration of **8** has already determined by the VCD spectrum of **8** and the crystal structure of **6**.

The reviewer suggested the single X-ray diffraction analysis of **8** in order to clarify the solid-state structure rather than to determine the absolute configuration. Did the authors try the crystallization of the mixture of (P,P)-**8** and (M,M)-**8**? General helicenes crystallize in the racemic form, in which (P)-helicene and (M)-helicene are often stacked in the crystal packing. Generally, racemic molecules are easy to crystalize as compared to the enantiomeric forms.

a) We thank the referee for recognising that the absolute configuration of **8** is correctly assigned based on our VCD work in conjunction with the X-ray structure of **6**. Regarding the solid-state structure of **8** from a racemate; although we believe that this would take away from the novelty of our enantiopure synthesis of this lemniscate we have, nonetheless, set up crystallisations from a racemic mixture of (P,P)-**8** and (M,M)-**8**. Sadly, we still have not obtained X-ray diffraction quality crystals. Despite this, we think the solid-state structure of *rac*-**8** is only relevant as an academic exercise and not to the F8BT films containing the enantiopure chiral dopant **8**.

b) Keywords in page 1, "Circularly Polarised Luminescence" is better.

b) Thank you for the suggestion. We have changed the keyword in the manuscript.

Reviewer #2 (Remarks to the Author):

Thanks for the authors' responses to all the reviewers. I appreciate the author have addressed some of my concerns. This reviewer fully agree that molecular modelling of F8BT doped thin films is challenging. On my opinion, however, the experimental glum values should be acquired for molecules **7** and **8** if they are luminescent. And relevant computations should be performed to disclose the mechanism of remarkable g values. I would like the authors to consider this comment.

We thank the referee for their comments. The glum graphs (and values) for both enantiomers of **7** and **8**, respectively have been included in the ESI (Figures S61 and S62). These figures show that neither **7** or **8** display any significant CPL in solution and therefore we cannot obtain glum values for the individual molecules. As such, computation of this would not be insightful.

In this manuscript we do not aim to elucidate the mechanism that leads to high glum values in F8BT thin films. We believe that this is a very complex question that requires the combination of extensive experimental and computational work which cannot be included in a communication. There is a lot of debate in the literature about this; here are a few papers where the origin of high glum values is discussed, but no consensus has been reached: Jpn. J. App. Phys. 63, 2024, 02SP10; Chirality 35, 2023, 817; Angew. Chem. Int. Ed. 61, 2022,

202203075; Nat. Commun., 13, 2022, 210; Nat. Commun. 11, 2020, 6137; Adv. Mater. 25, 2013, 2624; ACS Nano 2017, 11, 12713; J. Phys. Chem. A 2012, 116, 1121; J. Am. Chem. Soc. 2003, 125, 14032.

Needless to say, we would like to contribute to the understanding of this chirality induction and we are currently working towards this, but more time and resources are needed to complete this study. Such an investigation represents a standalone study which, if included in the current manuscript, would take away the focus of this chemistry describing the enantiopure lemniscate synthesis and its use in producing chiral thin films with exceptionally large glum values.

We would like to thank the reviewers for their work on our manuscript.

Reviewer #1 (Remarks to the Author):

The authors responded to the almost all the reviewer's comments, and the present manuscript has been improved. In my opinion, this manuscript can be accepted in Nature Communications in the present form.

We thank the referee for their positive comments.

Reviewer #2 (Remarks to the Author):

The authors' responses are appreciated. However, on the opinion of this reviewer, the computational g factors are possible for small molecules (7 and 8), and also should be acquired regardless of the experimental g values. It is exciting that the authors "would like to contribute to the understanding of this chirality induction and we are currently working towards this". Honestly, this reviewer believes a clear elucidation of mechanisms for high g values would be one of the critical contributions that make it possible to be published in top journals. At this point, I disagree with the authors' statement "Such an investigation represents a standalone study which, if included in the current manuscript, would take away the focus of this chemistry describing the enantiopure lemniscate synthesis". In the case of only giving an example of chiral thin films with large g values but without teaching people why you could do that (I am sure our readers are quite interested in this), I would suggest the publication in a more specific journal.

We thank the referee for their comments. We have calculated and included in the ESI the dissymmetry factors for molecules **6**, **7** and **8** (Table S2, page 31).

We hope that this article along with the follow-up work from our group will help elucidating how the chiral dopants induce chirality in conjugated emissive polymers.